# Cancer Vaccines and Oncolytic Viruses Exert Profoundly Lower Side Effects in Cancer Patients than Other Systemic Therapies: A Comparative Analysis

**DOI:** 10.3390/biomedicines8030061

**Published:** 2020-03-16

**Authors:** Volker Schirrmacher

**Affiliations:** Immune-Oncological Center Cologne (IOZK), D-50674 Cologne, Germany; V.Schirrmacher@web.de

**Keywords:** targeted therapy, immunotherapy, oncolytic virus, cancer vaccine, adverse events, small molecule inhibitor, CAR T cell, checkpoint inhibitor, monoclonal antibody

## Abstract

This review compares cytotoxic drugs, targeted therapies, and immunotherapies with regard to mechanisms and side effects. Targeted therapies relate to small molecule inhibitors. Immunotherapies include checkpoint inhibitory antibodies, chimeric antigen receptor (CAR) T-cells, cancer vaccines, and oncolytic viruses. All these therapeutic approaches fight systemic disease, be it micro-metastatic or metastatic. The analysis includes only studies with a proven therapeutic effect. A clear-cut difference is observed with regard to major adverse events (WHO grades 3–4). Such severe side effects are not observed with cancer vaccines/oncolytic viruses while they are seen with all the other systemic therapies. Reasons for this difference are discussed.

## 1. Introduction

Standard therapy for solid tumors such as breast cancer includes surgery, radiotherapy, chemotherapy, and anti-hormone therapy. The main object of surgery consists of resection of localized operable stages with curative intent. Radiotherapy exerts an adjuvant function in combination with R0 resection and an additive palliative function in combination with R1/2 resection. Chemotherapy is performed either pre-operative (neo-adjuvant) or post-operative (adjuvant). Anti-hormone therapy is applied with hormone receptor expressing tumors as adjuvant therapy and in combination with adjuvant chemotherapy [1,2,3,4].

Since the efficacy of these types of systemic therapy is rather low [5] and the side effects high, there is a great need to develop new concepts and types of treatment. Molecularly targeted therapy (TT) is one such new concept [6]. It attempts to interfere with signaling pathways [7] which play an important role in distinct types of cancer. It took several decades of basic research until the first small molecule inhibitor (SMI), imatinib mesylate (Gleevec), was developed. It was approved in 2002 for application in patients suffering from chronic myeloid leukemia (CML) and gastrointestinal stromal tumors (GIST) [8].

A change of paradigm in terms of cancer treatment gradually developed with the concept of immunotherapy [9,10] based on cellular and molecular immunology [11]. The new focus was directed towards the immune system of the tumor-bearing host rather than towards the tumor itself. Mechanisms of tumor–host interaction were discovered [12] and the tumor microenvironment (TME) [13] was found to play an important role in this context. The TME is a consequence of malignant transformation. A growing tumor creates a shield that protects its cells from destruction by immune and non-immune mechanisms. For more details about innate and adaptive immunity in the TME see [14].

Since the 1950s, basic research in immunology and virology paved the way for the development of new therapeutics, such as monoclonal antibodies (mAbs) [15], immune T cells [16], cancer vaccines [17], and oncolytic viruses (OVs) [18]. Several of these new reagents received approval by the US FDA based on positive results from years of clinical studies.

This review compares mechanisms of function and extent of side effects by selected examples of these diverse therapeutics. The analysis is based on specific textbooks to chemotherapy [19], small molecule inhibitors (SMIs) [6], molecular vaccines [20], cancer vaccines [21], viral oncology [22], and viral therapy of cancer [23]. In addition, it involved a search using Pubmed on the side effects of new therapeutics approved during the years 2000–2020.

## 2. Chemotherapy

### 2.1. Mechanism of Action

Cytostatic drugs [24] are toxic for normal and malignant proliferating cells. Cellular targets of approved cytostatic drugs depend on the type of drug. For alkylating agents (e.g., cyclophosphamide, busulfane, dacarbacine) these are DNA, purin nucleosides, glutathione, and proteins. For alkaloids, (e.g., vinca-alcacoid, epipodophyllotoxin, camptothecin), these are tubulin and topoisomerase I and II. For antibiotics (e.g., bleomycin, anthrachinons, actinomycin D, mitomycin C), cellular targets are DNA, topoisomerase II, and RNA polymerase. Antimetabolites (e.g., methotrexate, 6-mercaptopurin, 5-fluoruracil, hydroxy-urea) target dihydrofolate acid reductase, purin biosynthesis, thymidylate synthetase, and ribonucleotide reductase [25].

### 2.2. Side Effects

Table 1 list immediate signs of toxicity of approved cytostatic drugs. The data are based on a respective textbook [25]. Toxicity is quantified according to the WHO classification (grade 0–4). Particular attention is payed to toxicity of grade 3 (severe) and 4 (life threatening or disabling). From bone marrow and blood are given the values for hemoglobin (Hb) and the numbers of cells (leucocytes, granulocytes, thrombocytes) in the different categories. Toxicity levels in the gastro-intestinal (GI) tract are characterized by the ratio of two liver enzymes (GOT and GPT) and by the level of bilirubin. Toxicity to the kidney can be followed by an increase in urea, creatinine, and proteinuria. Fever is another sign caused by chemotherapy.

Table 2 lists other signs of toxicity of approved cytostatic drugs. This time only grade 3 and 4 are given. All organs of the body are affected, including such essential ones as heart, lung and brain.

There exist also chronic (long-term) effects of chemotherapy, which include the development of drug resistance, carcinogenicity, and infertility [18].

## 3. Targeted Therapy with Small Molecule Inhibitors (SMIs)

Molecularly targeted therapies (TT) aim at blocking signal transduction pathways which represent hallmarks of cancer [25]. For this, new technologies of genomics, proteomics and pharmacogenomics are required to design small molecule inhibitors (SMIs) with the aim of improving the efficacy of cancer treatment and diminishing toxicity [26]. Such therapy can be given orally and it is applied to a selected targeted population of patients.

### 3.1. Mechanism of Action

Signal transduction inhibitors block the activities of molecules that participate in signal transduction, the process by which a cell responds to signals from its environment. During this process, the signal is relayed within the cell through a series of biochemical reactions that ultimately produce the appropriate response of the cell. In some cancers, the malignant cells are stimulated from within the cell to divide continuously without being prompted to do so by external growth factors. Signal transduction inhibitors interfere with this inappropriate signaling. Ras/Raf/MEK/ERK and Ras/PI3K/PTEN/Akt/mTor pathways play key roles in the regulation of normal and malignant cell growth [27,28]. Aberrant activation of the PI3K-AKT-mTOR pathway is one of the most frequent occurrences in human cancer and plays an important role in tumorigenesis [26].

The mitogen-activated protein kinase (MAPK-MEK-ERK-BRAF) pathway is an intracellular signaling pathway regulating cell cycle and other functions. Its activated kinase cascade drives a serial phosphorylation of the mitogen-activated extracellular signal-regulated kinase (MEK) and ERK kinases that leads to cell proliferation and survival. A paradigmatic example of this activation is melanoma, where deregulation of the MAPK pathway is evident in over 90% of the cases. In about 50% of cases, this is due to BRAF^V600^ mutation [26].

The KIT protein, a receptor for stem cell factor (SCF), of the KIT pathway was among the first described as activated through genetic alterations (translocations, deletions, point mutations, amplifications). Such changes in the Kit gene can function as driver in specific tumor types (e.g., gastrointestinal stromal tumor, melanoma, thymic carcinoma) in humans. In one study in small-cell lung cancer (SCLC), patients, whose tumors expressed Kit survived for an average of only 71 days after diagnosis, while those, whose tumors lacked Kit expression survived an average of 288 days [29]. The nature of the driver mutations in tyrosine kinases enabled to guide the administration of inhibitors of KIT, such as imatinib, sunitinib and others in a clinical setting.

A whole plethora of SMIs have been developed by pharma companies and were approved by the FDA. The pathway PI3K-AKT-mTor is targeted, for example, by the SMIs buparlisib, pictilisib, everolimus, or temsirolimus. BRAF and MEK pathways are targeted by vemurafenib, dabrafenib, or trametinib. The pathway ALK is targeted by imatinib, sunitinib, nilotinib, or valatinib. MET pathway is targeted by tivatinib, cabozantinib, crizotinib, or foretinib. There are many types of cancer for which distinct SMIs have been approved by the FDA.

Improved diagnostics allow to identify subsets of patients of a certain cancer that match a therapy with distinct SMIs thus opening the new field of personalized medicine.

### 3.2. Challenges for SMI Research

There are still a number of weaknesses and challenges of SMIs that need to be solved in the future [26].

The challenges from the side of tumor cells include:feedback loops and cross talk that can compensate for targeted inhibition,selection for mutations leading to resistance,selection of kinase switch variants.

The challenges from the side of the patient include:normal tissue toxicity including the immune system and the blood clotting system,the appearance of cutaneous squamous carcinoma,identification of suitable patients,the potentiation of adverse events in the case of combination with other TTs or with chemotherapy.

### 3.3. Side Effects

Unfortunately, SMIs have also severe side effects. Some of these are listed in Table 3. The most frequent ones relate to the skin and mucus membranes. In addition, there are common and serious side effects such as bleeding, slow wound healing and heart damage. Common are also diarrhea and problems with the liver, such as hepatitis and elevated liver enzymes. Many side effects are similar to those of standard chemotherapeutic drugs.

Changes in the skin are rather frequent and not only found by drugs interfering with epithelial growth factor receptor (EGFR) mediated signaling. It appears that signaling pathways are not that tumor-specific as one had thought or would have liked. Problems may be related not only to a lack of tumor-specificity, but may also have to do with delivery, dosing, and timing.

Resistance can occur in at least two ways: the target itself can change through mutation so that the TT no longer interacts well with it, and/or the tumor finds a new pathway to achieve tumor growth that does not depend on the target. Perhaps, TTs work best in combination. A recent study revealed that using two drugs that target different parts of the cell signaling pathway that is altered in melanoma by the BRAF^V600E^ mutation slowed the development of resistance and disease progression to a greater extent than using just one TT [30].

The idea of using multiple targets to avoid resistance development seems rational. Since SMIs have unwanted side-effects, the use of multiple SMIs would, however, also create multiple side-effects. It may be unpredictable in which way the drug effects will interact, an unacceptable risk.

Scientists had expected that TTs would be less toxic than traditional chemotherapy drugs because cancer cells are more dependent on the selected targets than are normal cells. However, as an enormous number of clinical studies revealed, TTs with SMIs can have substantial side effects.

In 2013, 40 drugs were dropped from the global oncology pipeline [31]. 12 drugs failed in Phase III development. None of the pivotal trials incorporated molecular biomarkers for patient stratification. The largest number of drug terminations (50%) occurred in Phase I development. Nevertheless, cancer research today delivers new treatments to patients faster than ever. In just one year, the US FDA has approved 20 therapies for more than a dozen different types of cancer.

## 4. Cytokines

Interleukin-2 (Il-2) has been approved for metastatic melanoma and renal cell carcinoma. Interferon (IFN)-α2a was approved 1999 for non-specific immunostimulation in Chronic myeloid leukemia, Hairy cell leukemia and melanoma. IFN-α2b has been approved for multiple hematological and solid tumors [32].

Other cytokines such as GM-CSF, IL-12, IL-15, IL-21, IFN-γ, and TNFα were found to have generally high toxicities and poor pharmacokinetics [32].

## 5. Immunotherapy with Monoclonal Antibodies, Including Checkpoint Inhibitors

Monoclonal antibodies (mAbs) have a number of advantages in comparison to SMIs: higher specificity and affinity, longer target binding, less frequent application (weekly versus daily), and lower side effects (rash, allergy). Herceptin (trastuzumab), as an example, targets the extracellular ectodomain of a tyrosine kinase growth factor receptor (HER2) of breast cancer, while SMIs interact with intracellular receptor domains. Herceptin shows therapeutic activity against HER2 positive breast cancer. Therapeutic mAbs have also been developed against other growth factor receptors (e.g., HER1, VEGFR, PDGFRα). FDA approved indications for mAbs include carcinomas, sarcomas, neuroblastoma, myeloma, as well as B- and T-cell lymphoma. MAbs also revolutionized diagnostics and research applications.

The greatest success with the application of mAbs in cancer patients has been in recent years the use of checkpoint inhibitory antibodies. The novel class of immunotherapy was first approved in 2011.

### 5.1. Mechanism of Action

Immune checkpoint receptors are crucial molecules for the fine-tuning of immune responses [33,34]. Checkpoint receptors on T cells such as CTLA-4 or PD-1 mediate negative dampening signals to T cells to avoid the destructive effects of an excessive inflammatory response and autoimmune reactivity. Tumors use several mechanisms to avoid elimination by the immune system. One involves hijacking checkpoint pathways. Checkpoint blockade therapy utilizes mAbs to release the breaks from suppressed T cells, allowing them to become activated and to recover their antitumor activity [35,36].

Single-agent application of anti-CTLA-4 [30] and anti-PD1 was surprisingly effective and caused an improvement in overall survival (OS) in metastatic melanoma. Impressive clinical efficacy was also observed in metastatic kidney cancer and in non-small-cell-lung cancer (NSCLC)—all malignancies that frequently cause brain metastases [36,37].

New emerging mAbs have great potential for the systemic control of epithelial cancers such as lung cancer. Reported phase I trials of nivolumab, MK-3475, MEDI4736, and MPDL3280A, are demonstrating durable overall radiological response rates in the range of 20–25% in lung cancer. Dual checkpoint blockade strategies, such as those combining anti CTLA-4, anti-LAG-3, or anti-KIR, are being tested to increase the proportion and durability of tumor responses.

With immunotherapies, new mechanisms of action require adaptations in study design and statistical analysis. Also, clinical trial endpoints need refining, taking into account the possibility of delayed treatment effects.

### 5.2. Side Effects

Clinical effects of therapies have always to be compared with their side effects. This is not different with immunotherapy. The side effects of blocking the immune systems natural inhibitory mechanisms have manifested clinically as diarrhea, rash, and hepatitis. The symptoms of side effects caused by such new reagents have been termed immune-related adverse events (irAEs) [38].

Table 4 lists the major side effects. Of particular significance are, apart from general fatigue, the endocrine effects: hypophysitis, thyroid disease, and adrenal insufficiency. Acute interstitial nephritis is possibly related to the presence of autoreactive clonal T cells. Renal monitoring every two weeks for 3–6 months has been recommended [38,39,40,41].

The diagnosis, monitoring, and management of irAEs following administration of anti-PD-1 checkpoint inhibitors have been described [42]. The time of onset ranges from one to six months. The majority of events are reversible by the use of glucocorticoids, notably methylprednisolone or equivalents. In contrast to single agent application, combined application of anti-PD-1 and anti-CTLA-4 immune checkpoint blocking antibodies has considerable toxicities. In this case, gains in efficacy have to be balanced against higher frequency and severity of irAEs [43]. Recently, the evaluation of two dosing regiments for nivolumab and ipilimumab in patients with advanced melanoma revealed a significantly lower incidence of treatment-related grade 3-5 AEs with NIVO3 + IPI1 in comparison to the approved combination NIVO1 + IPI3 [44].

Although steroids can be used to treat irAEs, the associated immunosuppression may compromise the antitumor response. The management of checkpoint inhibitory toxicity is complex and requires collaboration with subspeciality colleagues. Biomarkers predictive of efficacy and toxicity are also important to identify patients accurately who will benefit from checkpoint blockade.

“Third-generation” novel combinations based upon the PD-1/PD-L1 “backbone” have a relatively favorable safety profile and efficacy compared to other checkpoint inhibitors [45,46].

## 6. Immunotherapy via Cancer Vaccines and Oncolytic Viruses

Within the growing field of immuno-oncology, cancer vaccines and oncolytic viruses are promising strategic approaches. They can be defined and classified as active immunotherapeutic interventions because their antineoplastic effects are based on initiating a novel or boosting an existing immune response against cancer cells. Tumor antigen (TA) presenting vaccines can be based on peptides, DNA, and dendritic cells (DCs) as antigen-presenting cells (APCs). Oncolytic viruses (OVs) refer to non-pathogenic viruses that specifically infect cancer cells and cause oncolysis, thereby initiating post-oncolytic anti-cancer immunity. Like other agents (e.g. some cytostatic drugs, ionizing radiation), OVs can be categorized as immunogenic cell death (ICD) inducers. ICD was recently defined by consensus guidelines [23]. ICD facilitates the recruitment of APCs, guides the interaction between APCs and dying cells, favors phagocytosis of dying cells, promotes the maturation of APCs, their migration and their cognate interaction with T cells (cross-priming). 

Current anticancer immunotherapies have recently been classified. One detailed review included more than 500 references [32]. Another recent review described emerging targets and combination therapies [47].

### 6.1. Cancer Vaccines: State-of-the-Art

The purpose of cancer vaccine application can be preventive or therapeutic.

Preventive vaccines: Two licensed cancer-preventive vaccines proved to be effective: Anti-hepatitis B virus (HBV) to prevent HBV-associated hepatocellular carcinoma and anti-human papilloma virus (HPV) to prevent HPV-associated carcinoma of cervix, pharynx and larynx.

Therapeutic vaccines: There are currently 2 FDA-approved therapeutic cancer vaccines: 1. Sipuleucel-T, approved 2010, for metastatic prostate cancer. Sipuleucel-T is a dendritic cell (DC) vaccine pulsed with a recombinant fusion protein composed of prostatic acid phosphatase (PAP) fused to granulocyte macrophage colony-stimulating factor (GM-CSF) [48]. The approval of this DC-based vaccine for therapeutic treatment of prostate cancer represents a significant progress. Unfortunately, the producer (Dendreon Co.) filed for bankruptcy in 2014. 2. HSPPC-96 (Oncophage, vitespen) is a heat shock protein-peptide complex cancer vaccine [49,50,51]. This is an individually produced vaccine that relies on all TAs binding to HSPs including patient-specific neoantigens. Although the concept is advantageous in that it does not rely on a single TA, the production is associated with considerable costs [49].

Many of the studies with cancer vaccines have to be considered as experimental. For instance, plasmid DNA vaccines. These are molecularly defined types of vaccine capable to induce anti-tumor immune responses [52]. An example is the DNA vaccine HPV-16E7. To increase its efficacy, non-replicating NDV has been used as an adjuvant. The combination enhanced antitumor activity through the induction of tumor necrosis factor related apoptosis inducing ligand (TRAIL) and through granzyme B expression [53]. Similarly, in an earlier study, the hemagglutinin-neuraminidase (HN) gene of NDV had been demonstrated to function as a powerful molecular adjuvant for DNA anti-tumor vaccination. Co-expression of HN with a tumor target antigen through bicistronic vectors ensured precise temporal and spatial co-delivery to direct anti-tumor immune responses, preferentially towards Th1 [54].

#### 6.1.1. Mechanism of Action

Cancer vaccines are applied to cancer patients for the purpose of active specific immunization. Since the immune system of the patient should be capable to respond, neoadjuvant [55] or post-operative application has a better chance to function in comparison to late-stage disease.

The aim of this immunization is to induce a tumor-specific T cell response. This consists of Th1 polarized CD4+ T helper cells, T1 polarized CD8+ cytotoxic T lymphocytes (CTLs) and of TA-reactive memory T cells (MTCs).

Active-specific immunization via cancer vaccines instructs the patient’s immune system about relevant tumor-associated antigens (TAs), in particular about individually specific tumor neoantigens [56]. Professional antigen-presenting cells such as DCs are often part of a cancer vaccine. They can be loaded with autologous tumor lysate [57], with defined peptides [55,58,59,60], with tumor-derived DNA [61] or RNA [62] and also with viruses [63]. Recombinant lentiviruses expressing multiple tumor-specific antigens including stem cell antigen-2 (Sca-2) have been used successfully to produce effective DC based vaccine [63].

Apart from TAs, cancer vaccines contain adjuvants to activate innate immunity and to cause co-stimulation of TA-specific T cells. Lack or insufficient co-stimulation of TA-specific T cells in cancer patients causes T cell anergy. Vaccines have to be constructed in such a way that they can overcome immune tolerance and the influence of naturally occurring regulatory T (Treg) cells. As an example, vaccination by pHN plasmid at the ear pinna site of mammary carcinoma bearing mice caused changes in the tumor microenvironment by increasing NK cell infiltration and decreasing infiltration by myeloid-derived suppressor cells (MDSCs) [64].

Active-specific vaccination can be considered as a training program so that priming of TA-specific T cells leads to association of TA recognition with immunological danger signals. Danger signals are received via pattern-recognition receptors (PRRs) of innate immunity cells upon interaction with ligands from pathogen-associated molecular patterns (PAMPs) or from damage-associated molecular patterns (DAMPs).

Cancer vaccine induced immune responses exhibit unique kinetics, can target tumor neoantigens, and induce an antigen cascade [65].

#### 6.1.2. Side Effects

Vaccination with cancer vaccines is usually well tolerated and goes without serious adverse events (AEs). For instance, TA peptide-based vaccines spare normal tissue and thus have only low toxic effects. Cancer vaccines against neoantigens possibly provide the best ratio between targeting tumors (specificity) while sparing normal tissue (toxicity) [56].

### 6.2. Oncolytic Viruses

Oncolytic virotherapy [23,66,67] represents an interesting new class of cancer immunotherapy. Intra-tumoral application is often favored because it circumvents possible problems associated with systemic application. In 2005, a recombinant adenovirus (H101, Oncorine^R^) has been approved in China for treatment of head-and-neck cancer in combination with chemotherapy [68]. In 2015, the US FDA approved T-VEC, a recombinant oncolytic herpes simplex virus type 1 (HSV-1) with GM-CSF as transgene [69].

#### 6.2.1. Mechanism of Action

Oncolytic viruses (OVs) selectively replicate in and kill tumor cells. The therapeutic efficacy of OVs was thought for a long time to depend mainly on direct viral oncolysis. Meanwhile, the capacity of OVs to induce immunogenic cell death (ICD) is considered as key for eliciting potent anti-tumor immunity. ICD includes immunogenic apoptosis, necrosis/necroptosis, pyroptosis, and autophagic cell death. Endoplasmic reticulum (ER) stress response, PAMPs, and DAMPs play important roles in OV-induced ICD [70].

Oncolytic NDV, a naturally occurring attenuated avian RNA virus, can serve as a paradigm. Its mechanism of ICD has been described in detail [71]. In addition, oncolytic NDV has the potential to break therapy resistance [72]. OV-induced immunogenic events can override impaired TA presentation and can promote appropriate T cell interaction with antigen-presenting cells (APCs). Reovirus is another naturally occurring oncolytic RNA virus which exploits altered Ras signaling pathways in cancers. Reovirus showed pre-clinical efficacy, replication competence, and a low toxicity profile in humans [73].

Recent studies demonstrated that genetically modified OVs, armed with a therapeutic gene, mediated clinical responses. FDA approved T-VEC was safe and, in advanced melanoma, resulted in a 10.8% complete response rate, significantly higher than systemic GM-CSF alone [70]. Other examples are poxvirus (Pexavec, TRICOM, TroVax) [74,75], and adenovirus (ONCOS-102) [76].

#### 6.2.2. Side Effects

The reported side effects of oncolytic paramyxoviruses (attenuated newcastle disease virus (NDV), measles virus, mumps virus and sendai virus) were grade 1 and 2, with the most common among them mild fever [77]. With poxviruses (vaccinia and fowlpox), toxicity never exceeded grade 2 [78]. Oncolytic armed adenoviruses demonstrated a satisfactory safety profile [79]. T-VEC was reported to mediate clinical responses with few severe side effects [70].

Loading an oncolytic virus (e.g., vesicular stomatitis virus (VSV)) onto CD8+ central memory T cells improved safety and efficacy compared to systemic virus administration [80].

### 6.3. Cancer Vaccines Modified by Oncolytic Newcastle Disease Virus

Cancer vaccines can be modified by virus infection in order to increase their immunogenicity. This has been exemplified with NDV. Two types of such vaccine were developed over many years: (i) ATV-NDV, an autologous tumor cell vaccine modified by infection with a lentogenic strain of NDV [81,82,83,84,85], and (ii) IO-VAC^R^, a dendritic cell (DC) vaccine modified by loading with oncolysate from patient-derived tumor cells infected with a mesogenic oncolytic strain of NDV [58].

#### 6.3.1. Mechanism of Action

In a pre-clinical animal tumor model, post-operative vaccination with ATV-NDV but not with ATV without virus infection caused long-term protective anti-tumor immunity [83]. ATV-NDV strongly augmented a tumor-specific T cell response as a result of CD4+ and CD8+ immune T cell interaction [84]. In addition to the response activation via T-T cooperation, NDV of the vaccine caused induction of type I interferons which play an important role in the generation of a CTL response [81].

In a clinical study, post-operative vaccination with ATV-NDV of primary glioblastoma multiforme (GBM) patients caused: (i) increase of skin anti-tumor delayed-type hypersensitivity reactivity, (ii) an increase in the numbers of cancer-reactive memory T cells in the peripheral blood, and (iii) an increase of CD8+ tumor-infiltrating lymphocytes (TILs) in recurrent brain tumor tissue [85].

NDV introduces two types of PAMPs into the tumor vaccine: (i) cytosolic viral 5’-triphosphate RNA which is recognized by RIG-I receptors [86] and (ii) cell membrane expressed hemagglutinin-neuraminidase (HN) protein which is recognized by NKp46 receptors on NK cells and leads to their activation [87]. Innate immunity activation also involved monocytes and macrophages. This led to upregulation of TRAIL [88] and to induction of nitric oxide (NO) [89].

With regard to the DC vaccine IO-VAC, there has been a sophisticated analysis of the effect of NDV infection of human DCs. Within 18 hrs of infection, the DCs became re-programmed into polarization towards DC1. This cellular antiviral response was uninhibited and was dictated by a choreographed cascade of transcription factors [90]. In a clinical study with cancer-reactive MTCs from the bone marrow (BM) of breast cancer patients, in vitro stimulation with an IO-VAC-like vaccine caused bi-directional stimulation of DCs and T cells involving pro-inflammatory cytokines (TNF-α, IL-2, IL-15, IFNα, IFN-γ) [91].

#### 6.3.2. Side Effects

In one study a comparison was performed between the vaccine ATV-NDV and an autologous cancer vaccine admixed with BCG [92,93]. The side effects of the latter were more severe, including ulcer formation [82]. The most common side effects of ATV-NDV and IO-VAC were transient mild fever. Altogether, the grades of side effects with vaccines combined with OVs are < 2. This means that the small side effects of vaccines and OVs are not additive when the two components are combined. Table 5 gives on overview of the reported side effects.

## 7. Chimeric Antigen Receptor (CAR) T-Cell Therapy

Another recently developed immunotherapy is chimeric antigen receptor (CAR) T-cell therapy: T cells become transfected with a genetic construct containing an extracellular antibody binding site with specificity for a TA and an intracellular signal-transmitting chain. CAR T-cell therapy makes use exclusively of the effector arm of the immune system and bypasses immune instruction (e.g., vaccination), immune processing via DCs and immune recognition via natural TA-specific TCRs.

The US FDA and the European EMA recently approved two anti-CD19 CAR-T-cell products, YESCARTA* (Axicabtagene Ciloleucel) and KYMRIAH* (Tisagenlecleucel), for the treatment of patients with CD19-positive malignancies, such as refractory diffuse large B cell lymphoma (DLBCL) or relapsed or refractory B-cell acute lymphoblastic leukemia (ALL) in children and young adults [94,95].

### 7.1. Mechansm of Action and Costs

Lentiviral vectors are used to transfer the CAR construct into blood-derived T cells. CAR T-cell therapy has considerable potential in the treatment of various types of malignancies. It is expected to have long-term survival benefits but so far long-term survival data are limited.

CAR T-cell therapy makes use exclusively of the effector arm of the immune system and bypasses immune instruction (e.g., vaccination), immune processing via DCs, and immune recognition via natural TA-specific TCRs.

Cost-effectiveness estimates ranged from $82,400 to $230,900 per quality-adjusted life-year (QALY) for public payers [96].

### 7.2. Side Effects

One side effect is immunogenicity of the product due to the presence of non-human sequences and residual viral proteins or other non-human proteins utilized during the gene editing step. Such immune responses have an impact on CAR T-cell expansion and persistence and thus on the overall safety and reproducibility [97].

The most common adverse events are cytokine release syndrome (CRS) and encephalopathy syndrome (neurotoxicity). In one study, including 75 patients, 73% experienced grade 3 or 4 irAEs. CRS occurred in 77% and neurotoxicity in 40% of patients [96]. A Meta-analysis of 997 tumor patients from 52 studies revealed that CAR T-cell therapy had a higher response rate, but also a higher side effect rate in hematological malignancies as compared to solid malignancies [98].

## 8. Combination of Checkpoint Blockade with Cancer Vaccines, Oncolytic Viruses and Adoptive T-Cell Therapy

Combining immunotherapies has been proposed as a concept to convert “cold” into “hot” tumors [99].

In a preclinical study, anti-PD-1 antibody therapy was found to potently enhance the eradication of established tumors by scFv-anti-Her-2 CAR T-cells [100]. Similarly, in another preclinical study, PD-1 blockade enhanced DC vaccine induced immune responses in glioma [101].

A phase Ib/II clinical study in non-small cell lung cancer (NSCLC) evaluated the combination of anti-PD1 (nivolumab) with an allogeneic lung tumor cell vaccine secreting Gp96-Ig HSP-peptide complexes and Fc-OX40L costimulatory molecules (Viagenpumatucel-L). In preclinical studies, this cell-based vaccine led to improved T-cell priming, increased immune stimulation without off-target consequences, increased memory T cell precursor cell (CD127+ KLRG-1-) production and tumor elimination [102]. Since the clinical study results are not yet published, no information on side effects can be given.

A randomized, open-label phase II study evaluated the efficacy and safety of the addition of an OV to a checkpoint inhibitor in patients with advanced, unresectable melanoma. The objective response rate was significantly higher in the combination arm (39% versus 18%). Unfortunately, the incidence of grade > 3 AEs was also higher in the combination arm (45% versus 35%, mostly fatigue, chills and diarrhea) [103].

Intralesional therapies are locoregional therapies and are likely to reduce side effects from systemic therapies. A multitude of intralesional immunotherapies have recently been evaluated in clinical trials. They included OVs (CAVATAK, Pexa-Vec, HF10), nononcolytic viral therapies (PV-10) and toll-like receptor 9 agonists. Such studies revealed promising antitumor activity with tolerable toxicities in melanoma and other solid tumors [104].

## 9. Discussion

A meta-analysis of newly approved anti-cancer drugs revealed not only high costs [105], but also high risks of associated toxicities [106]. It was questioned whether this is the price we have to pay for progress [106]. The design and reporting of many randomized controlled trials (RCTs) can render their results of little relevance to clinical practice. Suggestions have been made how to improve the clinical relevance of RCTs [107]. This review has compared different types of anti-cancer drugs and reports that some of them have profoundly lower side effects than others. The mechanisms of the various types of systemic therapy are explained and will be discussed.

### 9.1. Chemotherapy

Excessive cell proliferation is a characteristic of cancer. It is therefore not surprising that the oldest types of systemic therapy, radio- and chemotherapy, are directed against cell proliferation. Direct effects of ionizing irradiation on cancer cells include DNA and chromosomal damage, mitotic catastrophe, senescence and apoptosis induction. Chemotherapy became established in the 1970s as treatment for advanced Hodgkin’s disease, non-Hodgkin’s lymphoma, teratoma of testis and as adjuvant treatment for early breast cancer. Unfortunately, chemotherapy cannot be considered as a curative treatment procedure for the most common cancers, the carcinomas. In addition, chemotherapy exerts unwanted toxic activity towards normal proliferating cells within body tissues such as bone marrow, endothelia, and hair follicles. It also exerts negative effects on the immune system.

Another shortcoming of chemotherapy is that it does not affect tumor stem cells which are non-proliferating and quiescent. The cell cycle is a precisely programmed series of events divided into four phases: G1 (Gap 1), S (synthesis), G2 (Gap 2), and M (mitosis). G0 or quiescence occurs when cells exit the cell cycle. The cell cycle clock which controls the events from G1 to M operate similarly in all cell types throughout the body. This is a major explanation for the enormous adverse events of chemotherapy.

### 9.2. Molecularly Targeted Therapy

The discovery of cellular oncogenes and tumor suppressor genes in the 1970s and 1980s were milestones in molecular biology research of cancer and paved the way for a new type of cancer therapy, molecularly targeted therapy (TT). TT was considered as break-through in cancer treatment because it had a scientific and rational basis: Research in tumor virology led to the discovery of a number of viruses which apparently had incorporated cellular oncogenes. Elucidation of the mechanism of function revealed first clues about cell signaling via growth factors and corresponding receptors and their importance for tumor growth. Examples are *Rous sarcoma virus* carrying the v-src oncogene, avian erythroblastosis virus carrying the v-erbB oncogene, *Simian sarcoma virus* carrying the v-sis oncogene and Friend leukemia virus carrying the gp55 env gene coding for a mimic of the growth factor erythropoietin (EPO). A new theory arose, namely that a cellular growth factor gene or a growth factor receptor gene could have been hijacked by a virus to become an oncogene. Gradually it became clear that growth factors and their distinct tyrosine kinase receptors (TK-Rs) are often involved in tumor pathogenesis. Pharma companies thus engaged in the development of small molecule inhibitors.

Meanwhile, SMIs have been developed that target epidermal or fibroblast growth factor receptor pathways. Others target apoptosis pathways, androgen pathways or vascular endothelial growth factor (VEGF)-mediated angiogenesis pathways. Some SMIs can also target stem cells, DNA repair, or mitosis. It was expected that TTs would be less toxic than cytostatic drugs because cancer cells are more dependent on the selected targets than are normal cells. However, as an enormous number of clinical studies revealed, TTs with SMIs can have substantial side effects.

Apart from the side effects and the challenges for SMI research mentioned in Section 3.2, there exist further problems: For instance, some human cancers produce as many as three distinct growth factors (e.g., tumor growth factor α, stem cell factor, insulin-like growth factor) and at the same time express the receptors for these ligands, thus establishing three autocrine signaling loops simultaneously. The application of a corresponding number of SMIs appears highly problematic due to unknown drug interactions and the multiplication of side effects.

### 9.3. Immunotherapy

Immunotherapy is a strategy which involves the patient’s immune system to fight cancer. The immune system avoids attacking the body, maintains its integrity and retains a memory of successful defenses. Tolerance mechanisms within the immune system are important to understand the low side effects of immunotherapies. The types of immunotherapy selected for this review are based on T-cell mediated immunity.

### 9.4. Checkpoint-Inhibitory Antibodies

Checkpoint-inhibitory antibodies interfere with tumor immune escape mechanisms which deliver negative signals to activated T cells. The application of such antibodies resulted in an improvement of long-term survival in a significant proportion of patients. This suggests that cancer-reactive T cells had been produced in these patients spontaneously so that the release of the tumor-induced breaks revealed their therapeutic potential. The success of these new therapeutics in a clinical setting corroborates the concept of immune surveillance.

Since checkpoint-inhibitory antibodies interfere with immune regulation, it does not come as a surprise that they also induce immune-related adverse events such as auto-immune phenomena (Table 4). Early recognition and quick interventions are necessary and make the treatment somewhat demanding for the clinic.

### 9.5. CAR T-Cell Therapy

CAR T-cell technologies, although having considerable therapeutic potential, are also faced with severe toxicity problems (irAEs of 3–4). To reduce these, new strategies aim at introducing inducible gene switches [108,109].

Toxicities of novel therapies, such as checkpoint inhibitors, tyrosine kinase inhibitors and CAR T-cell therapies necessitate management and prevention strategies. A recent review addresses problems associated with the accelerating speed of new drug approval by the FDA and points to the challenge of management of real-world toxicity after drug approval. According to the authors, the broad spectrum of new side effects require special alertness [110].

### 9.6. Cancer Vaccines and OVs

Major adverse events are not a problem with immunotherapies involving cancer vaccines and/or oncolytic viruses. OVs are tumor-selective agents causing immunogenic cell death, thereby stimulating adaptive anti-tumor immune responses. Cancer vaccines instruct the immune system about tumor antigens and provide T-cell co-stimulatory signals. The paradime of maximal tolerated dose (MTD) developed with cytostatic drugs does not apply to cancer vaccines and OVs. Greater cytotoxicity by high doses of OVs does not neccessarily coincide with optimal immunogenicity [23].

It is often questioned whether therapeutics which hardly produce side effects can be effective. The references of Table 5 provide examples for immunotherapeutic activity in absence of adverse events. For instance, in a randomized trial, neoadjuvant vaccination with HER2 peptide-pulsed DC1 cells caused 28.6% pathologic complete responses in HER2+ ductal breast carcinoma in situ (DCIS) patients [55]. The low side effects are a consequence of the high specificity of the immune responses and of tolerance mechanisms within the immune system which avoid anti-self-reactivity. High specificity and effectivity is observed in particular with T cell responses against tumor neoantigens. Such antigens can be produced in cancer cells as a result of mutational or other genetic events. They are not affected by self-tolerance mechanisms.

Since it is important to understand the functioning of immune therapies with low side effects, this paragraph presents some major characteristics. Neoantigen-derived peptides are recognized by T cells as non-self peptides in association with self MHC molecules. This holds true for CD4+ T helper cells and CD8+ CTLs. These two types of T cells recognize different peptides from the neoantigen in association with MHC class II or MHC class I molecules, respectively. CD4–CD8 T–T cell cooperation during the response is a prerequisite for an effective anti-tumor response. This results in the generation of long-lasting neoantigen-specific memory T cells. Cognate interactions between antigen-specific T cells and APCs involve APC scanning, immunological synapse formation, bidirectional cell stimulation, T-APC cluster formation, generation of T lymphoblasts and clonal T cell expansion [111,112,113]. Such processes can occur at different sites, e.g., in lymph nodes, spleen or bone marrow and also at the site of the tumor. CTLs kill tumor cells via a lethal hit signal at the cytotoxic secretory synapse via unidirectional perforin pore delivery. Multiple tumor target cells are killed per CTL via exocytosis of cytotoxic granules which are recycled in target cells [114].

Research in virology also contributed to the development of new cancer therapeutics. Oncolytic viruses emerged as promising anticancer reagents. OVs have been developed from various virus families, such as *Herpesviruses, Adenoviruses, Paramyxoviruses, Rhabdoviruses, Poxviruses,* or *Retroviruses*. In 2015, the first OV (T-VEC, *HSV* encoding GM-CSF) became approved by the FDA for melanoma immunotherapy. OVs selectively replicate in tumor cells and lyse them. After this direct oncolysis phase, a post-oncolytic immune response continues to mediate tumor destruction. This associated immune response is due to virus-induced immunogenic cell death involving immunogenic apoptosis, necroptosis and pyroptosis. Often, the viruses need to be genetically modified in order to achieve tumor selectivity, oncolysis, safety and low toxicity.

A further development represents the combination between a cancer vaccine and an OV. Innate immunity activation via an OV provides costimulatory signals for activation of T cells specific for TAs of the vaccine. One strategy used NDV-infected autologous tumor cells as vaccine thus combining high specificity of TA expression with the co-stimulatory activity of NDV. In a prospectively randomized clinical trial in stage IV colon cancer patients, post-operative vaccination with ATV-NDV caused significant long-term (>10 years) improvement of OS [115]. The side effects upon combining cancer vaccine and OV were mild, similar to those of each agent alone.

In contrast to chemotherapy, immunotherapy by vaccines and OVs shows virtually no toxicity towards normal proliferating cells. In addition, immunotherapy has the potential to kill non-proliferating tumor cells such as tumor stem cells, while such cells are resistant to chemotherapy.

There is at present a trend towards personalized instead of standardized medicine. While treatment with cytostatic drugs has become standardized, the treatment with SMIs is personalized, i.e., adapted to the genetics and other characteristics of the patient’s tumor. The immunotherapy with autologous virus-modified vaccines goes one step further in being individualized. The therapeutic is produced for each patient individually. This is because tumor neoantigens are individually unique.

### 9.7. Adoptive T-Cell Therapy

The potential power of immunotherapy goes far beyond what is seen with checkpoint inhibitory antibodies. This potential can only be exploited to its full extent based on innovative basic and applied immunological research. An example as to what immunotherapy can achieve in a graft-versus-leukemia animal model: A single transfer of tumor-reactive immune T cells from an MHC-matched allogeneic mouse strain into a recipient strain with advanced metastasized disease caused complete reversion of cancer-induced dysregulation, including: (i) primary tumor targeting, encapsulation, and rejection, (ii) eradication of liver metastases, (iii) reversion of cachexia. GvH-mediated damage of the liver induced recruitment of mesenchymal stem cells from the bone marrow which then supported liver regeneration [116].

The transferred T cells had been pre-immunized against the tumor and were equipped with high specificity and diversity (directed against TAs, minor histocompatibility antigens and against a viral superantigen). The identified weapons of the donor T cells and of host-derived APCs were granzyme B, perforin, TRAIL, and NO [117].

## 10. Conclusions

Cancer treatments by means of chemicals such as cytostatic drugs or small molecule inhibitors are often associated with severe side effects. Cancer treatment by immunotherapy based on vaccines and oncolytic viruses is much better tolerated. There are a number of differences between these types of treatments.

Since T cells are mobile, T cell mediated immune responses are systemic. They are self-regulatory and involve an expansion and a retraction phase. Cognate interactions between APCs and T cells are guided by chemokines. Double recognition of TA-derived peptides via MHC class I and class II molecules provides the basis for T-T cell interaction. Self-tolerance mechanisms and the requirement for co-stimulatory signals ensure that the immune response does not become directed against healthy normal cells. Finally, after the retraction phase, a small proportion of the antigen-specific T cells survive with a memory function. Thus, the immune system functions as an integrated unit with a number of different cell types participating and stimulating or controlling each other.

Such biological functions are lacking when it comes to cytostatic drugs or even to targeted small molecules.

Oncolytic viruses are tumor selective self-amplifying cancer toxic biological agents. If they harbour a therapeutic transgene, this will be amplified as well during the viral replication cycle. OVs cause ICD and thus also have a positive effect on the anti-tumor immune response. The combination of cancer vaccines with OVs has a rational basis, namely augmentation of the anti-tumor response without increasing side-effects.

The review is a plea for further immunotherapy, lower side effects, and higher efficacy with respect to cancer treatment.

## Figures and Tables

**Table 1 biomedicines-08-00061-t001:** Toxicity of approved cytostatic drugs according to World Health Organization (WHO) classification.

Grade	0	1	2	3	4
Intensity	none	mild	moderate	severe	life-threatening or disabling
1. Bone marrow & blood ^a^					
Hb	> 11	10–11	8–9	7–8	< 6.5
Leucocytes	> 4	3–4	2–3	1–2	< 1.0
Granulocytes	> 2	1–2	1	0.5–0.9	< 0.5
Thrombocytes	> 100	75–99	50–74	25–49	< 25
2. GI Tract					
Liver GOT/GPT	< 1.25 × N ^b^	1.25–2,5 × N	2.6–5 × N	5.1–10 × N	> 10 × N
Bilirubin	< 1.25 × N	1.25–2.5 × N	2.6–5 × N	5.1–10 × N	>10 × N
3. Kidney					
Urea	< 1.25 × N ^b^	1.25–2.5 × N	2.6–5 × N	5.1–10 × N	> 10 × N
Kreatinin	< 1.25 × N	1.25–2.5 × N	2.6–5 × N	5.1–10 × N	> 10 × N
Proteinuria	none	< 3 g/L	3–10 g/L	> 10 g/L	ne syn
4. Fever ^d^	none	< 38 °C	38–40 °C	> 40 °C	pr dec

^a^ Hb = hemoglobin (g/100 mL); cells (× 10^9^/L); ^b^ N = norm value; ^c^ ne syn = nephrotic syndrome; chemotherapy (Mit. C) induced hemolytic-uremic syndrome (c-HUS) with letality between 44 and 82%; ^d^ pr dec = fever caused by therapy and not by the tumor, combined with blood pressure decrease (hypotony). GI = gastrointestinal tract; GOT = glutamate-oxalacetate-transaminase; GPT = glutamate-pyruvate-transaminase.

**Table 2 biomedicines-08-00061-t002:** Other signs of toxicity of approved cytostatic drugs according to WHO classification.

Grade	3	4
Stomatitis	ulcers	peroral nutrition impossible
Diarrhoe	intolerable	hemorrhagic dehydration
Obstipation	subileus	Ilius
Hematuria	macrohematuria	obstructive uropathy
Lung	dyspnoe	bed stay obligatory
Allergy	bronchospasms	anaphylaxis
Skin	ulcerations	dermatitis, necrosis
Hair	alopecia, reversible	alopecia, irreversible
Infections	severe	severe + hypotonia
Heartfunction	dysfunction	dysfunction + nonresponsive
Bleading	severe	circulatory disorder
**Neurotoxicity**		
(i) central/consciousness	somnolencia >50%	coma
(ii) peripheral	paresthesia	paralysis
(iii) extrapyramidal symptoms	ataxia > 4 days	spasms, coma

**Table 3 biomedicines-08-00061-t003:** Possible side effects from targeted small molecule inhibitory drugs.

Grade 2–4
**Skin**
Changes in how the skin feels
Increase of photosensitivity
Rash (scalp, face, neck, chest, upper back
Dry skin
Itching
Red, sore cuticles (the areas around the nails)
Hand-foot syndrome, painful
Changes in hair growth
Changes in hair or skin color
Changes in and around the eyes
**Common and serious side effects**
High blood pressure
Bleeding or blood clotting problems
Slow wound healing
Heart damage
Swelling
Diarrhea
Hepatitis

Other side effects are similar to those of standard chemotherapy drugs.

**Table 4 biomedicines-08-00061-t004:** Side effects of immune checkpoint inhibiting mAbs: Immune-related adverse events (irAEs).

Grade 2–4
Skin: rash, rarely bullous pemphigoid (BP) Lung: Pulmonary toxicity, pneumonia Heart: autoimmune myocarditis, cardiovascular toxic effects Liver: hepatitis Gastrointestinal: diarrhea, vomiting, colitis (all grades and high grade) Kidney: acute interstitial nephritis Endocrine: hypophysitis, more rarely thyroid disease, occasionally adrenal insufficiency Fatigue: Fewer high grade events with anti-PD-1 than with anti-CTLA-4 mAbs

**Table 5 biomedicines-08-00061-t005:** Side effects of cancer vaccines and of oncolytic viruses.

Grade	0	1	2	3	4	References ^1^
Cancer vaccines	+	+	+/−	−	−	[47,48,49,50,51,52,53,54,55,56,57,58,59,60,61,62,63,64,65]
Oncolytic viruses	+	+	+/−	−	−	[22,23,66,67,68,69,70,71,72,73,74,75,76,77,78,79,80]
Virus-modified						
cancer vaccines	+	+	+/−	−	−	[57,71,72,73,74,75,81,82,83,84,85]

^1^ References refer to studies which report on therapeutic effects and on side effects; no side effects were observed of grade 3 or 4.

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
