# Peer review of "Cancer Vaccines and Oncolytic Viruses Exert Profoundly Lower Side Effects in Cancer Patients than Other Systemic Therapies: A Comparative Analysis"

_biomedicines, 2020, doi:10.3390/biomedicines8030061_

Round 1

Reviewer 1 Report

This article is written well, but lacks in new knowledge and grounds. Therefore, I require Major Revision.   Major point This study has too little data to discuss 'Cancer vaccines and oncolytic viruses'. Therefore, I require Major Revision. ・Please describe the analysis method of this study in detail in material and methods section. ・First of all, the definition of 'Cancer vaccines and oncolytic viruses' should be clearly understood. ・There are too few meaningful references number for academic arguments. In this study, please respond by increasing the number of appropriate references. ・This paper has not been analyzed and discussion of the problem. ・And, please describe the discussion of the mechanism.   Mainor point ・The sentence of this paper has many careful mention errors. Please review it.

Author Response

Major revisions were made as follows:

-The Basis of the Analysis is mentioned (end of Introduction)

  • A new Paragraph (4.) About cytokines has been added
  • In Paragraph 6. cancer vaccines and oncolytic viruses are first defined
  • The paragraphes About approved therapeutic vaccines and approved oncolytic viruses has been revised
  • A new Paragraph (8.) is added About combination therapies
  • The first Paragraph of the Discussion has been revised and extended
  • 20 new references are added
  • The tables have been revised

Reviewer 2 Report

The article “Cancer vaccines and oncolytic viruses exert profoundly lower side effects in cancer patients than other systemic therapies: A comparative analysis” is a review in which the author compares the side effects of different cancer therapies, from radiation and chemotherapy to targeted biological therapies, finishing with immunotherapies and oncolytic viruses. The author emphasizes on the fact that cancer vaccines and oncolytic viruses show less severe adverse effects in contrast to other therapies.

The article is in general well written and includes several tables where side effects of each type of therapy are listed. However, to be a bit more complete the author should include some additional information about the toxicity of the combination of Oncolytic viruses with immune checkpoints inhibitors, since this type of combination has also shown very good results in clinical trials.

Some addicitional minor points:

- The tables cannot be seen properly, with some columns being displaced

- Table 2, Page 4, line 5, Is the word “somnolencia” correct in Ennglish? Maybe it should be written drowsiness

- Table 3. Are all the side effects listed grade 2? (they seem to be under the 2 column)

- Part 5.1. I do not think BCG or T-VEC can be really considered as vaccines since they do not encode any antigen… In particluar T-VEC should be moved to part 5.2 which is dedicated to OVs

Page 9, line 47. NDV was previously defined…

Page 10, line 18. What is T-T cooperation? Please define it.

Author Response

All minor points are corrected

A new Paragraph (8.) has been added concerning the combination of checkpoint Inhibitors with other therapies

Also, a new Paragraph (4.) has been added with regard to cytokines

The tables have been revised

20 new references have been added

Round 2

Reviewer 1 Report

Cancer vaccines and oncolytic viruses exert profoundly lower side effects in cancer patients than other systemic therapies: A comparative analysis

This paper has been greatly improved according to my proposal.